REGISTERED REPORT PROTOCOL

# Semantic priming supports infants' ability to learn names of unseen objects

**Elena Luchkina**[1,2]* , **Sandra R. Waxman**[1,2]

**1** Department of Psychology, Northwestern University, Evanston, IL, United States of America, **2** Institute of Policy Research, Northwestern University, Evanston, IL, United States of America

☯ These authors contributed equally to this work.
* elena.luchkina@northwestern.edu

**Data Availability Statement:** We agree to make our raw data, any digital study materials, and

## Abstract

Human language permits us to call to mind objects, events, and ideas that we cannot witness directly. This capacity rests upon abstract verbal reference: the appreciation that words are linked to mental representations that can be established, retrieved and modified, even when the entities to which a word refers is perceptually unavailable. Although establishing verbal reference is a pivotal achievement, questions concerning its developmental origins remain. To address this gap, we investigate infants' ability to establish a representation of an object, hidden from view, from language input alone. In two experiments, 15-month-olds ($N = 72$) and 12-month-olds ($N = 72$) watch as an actor names three familiar, visible objects; she then provides a novel name for a fourth, hidden fully from infants' view. In the Semantic Priming condition, the visible familiar objects all belong to the same semantic neighborhood (e.g., apple, banana, orange). In the No Priming condition, the objects are drawn from different semantic neighborhoods (e.g., apple, shoe, car). At test infants view two objects. If infants can use the naming information alone to identify the likely referent, then infants in the Semantic Priming, but not in the No Priming condition, will successfully infer the referent of the fourth (hidden) object. Brief summary of results here. Implications for the development of abstract verbal reference will be discussed.

## Introduction

Language is among our most powerful tools for learning and communication. It permits us to learn information that does not, or cannot, manifest perceptually at the time of learning [1], such as historic facts, hypothetical scenarios, or scientific constructs. From toddlers hearing words for absent and often unknown objects (e.g., "Daddy outside is fixing the *trellis*") to physicists conversing about gravity, the communicative power of language enables us to transmit new information across space and time. This capacity rests upon an appreciation that language can convey information that is decoupled from the here-and-now. This uniquely human power is enabled by *abstract verbal reference*–an appreciation that words are linked to mental representations that can be established, retrieved and modified, even when the entities to which they refer are not perceptually available [2–4]. How, and how early, do infants gain this appreciation?

analysis code available on the Open Science Framework (https://osf.io/).

**Funding:** Pilot data collection was funded by a National Institutes of Health grant #R01HD083310 awarded to SW. Any future funding sources that will be used to carry out the proposed research will be disclosed and acknowledged at the moment of the full manuscript submission. The funders had and will not have a role in study design, data collection and analysis, decision to publish, or preparation of the manuscript

**Competing interests:** The authors have declared that no competing interests exist.

Evidence documenting infants' ability to comprehend *displaced (or fragmented) reference* (an operational definition of abstract verbal reference, which entails reference to objects or events that were recently visible but subsequently removed from infants' view) provides valuable insight into these questions. A nascent appreciation of displaced, or fragmented, reference appears to be present by 12 months of age. Infants interpret pointing gestures as referential [5,6] and refer to a recently hidden entity by pointing to its previous location [7–12]. In addition, 12-month-olds understand that words can refer to recently hidden objects [13–15]. By 14 months, infants bring greater social knowledge to bear, responding to a speaker's verbal request for an absent object by selecting an object that a speaker prefers [16]. Infants at 14 months also use words to retrieve memories of past events [17]. They remember properties of named, but absent, objects [18] and produce communicative behaviors towards the location of recently hidden objects [19].

In each of these preceding instances, 12-to-14-month-olds' resolution to the puzzle of displaced reference hinges upon their use of recent visual "anchors"–reminders that scaffold their representation of the now-absent referent [20]. By 16 months, infants go further, successfully resolving displaced reference on the basis of verbal testimony alone [21], even if the absent entity has not previously been "anchored" within the context of the experiment [22]. However, in each of these studies, infants can rely on existing knowledge, acquired when those entities *were* visible.

What remains unknown is when infants successfully acquire *new representations and update them on the basis of language alone.* This is the signature of abstract verbal reference. We propose that if infants can establish a mental representation for the meaning of a novel word from language alone and later draw on that representation to identify a candidate referent of that novel word, this will constitute clear evidence for abstract verbal reference.

There is considerable evidence that 19- to 24-month-old infants resolve abstract verbal reference and successfully infer the meanings of novel words on the basis of linguistic information alone, in the absence of any visual referents [23–26]. For example, Ferguson et al. [2014; 27], showed that 19-month-old infants successfully identify the referent of a novel word even if that referent is not available when the name is introduced. In their study, 15- and 19-month-olds were presented with novel nouns that were arguments of known verbs, either animacy-selecting ("The dax is crying") or animacy-neutral predicates ("The dax is right there"). When the novel word was presented in linguistic constructions that specified animate arguments, 19-month-olds (but not 15-month-olds) successfully mapped the novel word to an animate object when it later became visible.

But why did 15-month-olds fail? The reviewed literature suggests that between 14 and 16 months, infants can resolve reference to absent objects. On possibility is they cannot yet establish new mental representations for novel words based on linguistic information alone. However, it is also possible that they do have the requisite linguistic and conceptual capacities, but that their relatively sparse lexical knowledge masks them (on average, infants in Ferguson et al. comprehended 4.3 verbs from the MacArthur Level II Short Form). For example, in Ferguson's task, infants were required to infer the meaning of a novel noun based on a known verb. Infants' verb knowledge at 15 months may have been too sparse to support the acquisition of novel noun arguments.

We therefore introduce a new paradigm that taps into infants' capacity for abstract reference, but relies instead on the nouns they know [on average, 15-month-olds comprehend 71 nouns; 28] and their emerging sensitivity to semantic neighborhoods [29,30]. We ask whether priming a semantic neighborhood supports infants' ability to establish a representation for the referent of a *novel* noun, even when the referent cannot be seen. At issue is [1] how early infants successfully resolve abstract verbal reference, as measured by their ability to acquire

novel word meanings from language alone, and [2] whether semantic priming is supports such learning.

## Experiment 1

The goal is to assess whether 15- to 16-month-olds can successfully establish a representation of a novel word's meaning from language input alone, and recruit this representation later to identify a referent of the novel word. We focused on infants aged 15–16 months because although there is ample evidence that they successfully resolve displaced reference for known words [16,18,22,27,31], there is no evidence that they learn novel word meanings without a referent being present.

### Method

The proposed work conforms with the US Federal Policy for the Protection of Human Subjects and Northwestern University Human Research Protection Program requirements. All research personnel have completed the required responsible conduct of research training and has been certified by the Collaborative Institutional Training Initiative. A protocol amendment that extends the scope of the protocol to include the proposed procedure was submitted and approved by the Institutional Review Board of Northwestern University. This research will be conducted based on caregivers' written consent and participants' willingness to complete the experiments. All participants will receive compensation, regardless of successfully completing the experiments.

**Participants.**   Seventy-two ($N$ = 24 per condition) infants from the Chicago area will be recruited from the laboratory database. Infants will have no more than 15% exposure languages other than English. Caregivers will complete a MacArthur Short Form Vocabulary Checklist: Level II [Form A; 32], augmented with words used in the experiments but not included in the form (see S1 Appendix). They will also complete a demographic questionnaire about their education, employment, gender, and race (see S2 Appendix).

The number of participants was determined based on power analysis performed with G*Power 3.1 [33] for a repeated measures between-subject F-test with target effect size $f^2$ = .1225 (medium; [34], *alpha* = .05, power = .8, number of groups (experimental conditions) = 3, number of trials = 4, and projected correlation between measurements = .5. We reasoned that compelling evidence would require at least a medium effect size of experimental condition. Based on these parameters, the required number of participants is 72. The planned analyses include Generalized Linear Mixed Model rather than an F-test. We used F-test-based effect size estimations for simplicity. The four test trials were treated as repeated measures.

Participants may be excluded and replaced due to fussiness, crying, or equipment failure. Further, because our Covid-19 Contingency Plan (see below) involves testing infants using Lookit–an online platform designed for developmental research–we expect a data loss rate of about 47–50% [35] due to infants' crying or fussiness, parental interference, or technological issues. This, we anticipate that the total number of infants recruited could reach 140.

**Timeline.**   We intend to begin data collection as soon as the current registered report is granted in-principle acceptance and project that all data will be collected and analyzed within a calendar year from that date. If the results of the Stage 1 review warrant major revision of the proposed experimental procedures, we intend to resubmit the report for Stage 1 review within one month after receiving reviewers' feedback.

**COVID-19 contingency plan.**   The global COVID-19 pandemic is currently preventing us from testing participants in the lab. We are transitioning the experiments in this report to Lookit–an online platform for developmental research. Using this platform, we will be able to

record participants' eye movements using web-cameras in their desktop or laptop computers and subsequently code their looking behavior manually. While the reportedly higher data loss rate may slow our data collection, Lookit's access to a larger sample of participants and at-home participation is likely to offset its associated data loss.

**Apparatus.** A Tobii T60XL corneal-reflection eyetracker will be used for stimulus presentation and data collection. The eyetracker has a sampling rate of 60 Hz, and a display size of 57.3 × 45 cm.

**Stimuli.** Infants will view a series of four video-taped vignettes, each featuring a new novel word-object pairing. Each video begins with an actor pointing three familiar objects, naming each with its familiar basic-level noun. Next the actor looks toward an object hidden behind her back, naming it with a novel word. See Fig 1 for a representation of a single trial and S3 Appendix for a complete list of stimuli. All stimuli will be made available on Open Science Framework.

**Stimulus selection and design.** We used vocabulary norms [36] to select the visual and linguistic materials. For the familiar objects presented in the Priming phase, we selected objects whose names are understood by at least 60% of 15-month-olds (nouns: apple, banana, orange, jacket, sock, hat, truck, car, bus, cat, dog, horse). For the novel objects presented during Test, we (a) selected objects whose names infants do not understand (e.g., rare tropical fruits, exotic animals), and (b) chose pairs of objects from distinct semantic neighborhoods (e.g., papaya vs aardvark).

**Procedure.** All infants will participate in 4 trials, each including a Priming and a Test phase (Fig 1). Each trial will include one novel object-novel word pairing. Infants will be assigned randomly to condition. What varies across conditions is the first three objects presented during Priming: in the Semantic Priming and Follow-up Control conditions, these are all members of the same semantic neighborhood (e.g., animals, food, clothing); in the No Priming condition, they are from different semantic neighborhoods. We selected a between-subject design to avoid potential carry-over effects, which would dilute the predicted advantage of Semantic Priming. Another solution to minimize carry-over effects might be to increase the number of trials and to present them as blocks. However, doing so would lengthen the procedure and therefore runs the risk of exceeding infants' attention. In all conditions we provided infants with rich communicative and referential cues to support their interpretation of Trial 4 of the Priming phase, in which the object is hidden, as a labeling episode as well.

Semantic Priming condition ($N = 24$; Fig 1): Priming (35 s): During Priming an actor will point to and name three familiar objects, all from the same semantic neighborhood (fruits, clothing, vehicles, animals). The objects will appear one at a time, behind the actor; the actor will turn to point to and name them, using their familiar basic-level nouns (e.g., "Look! An

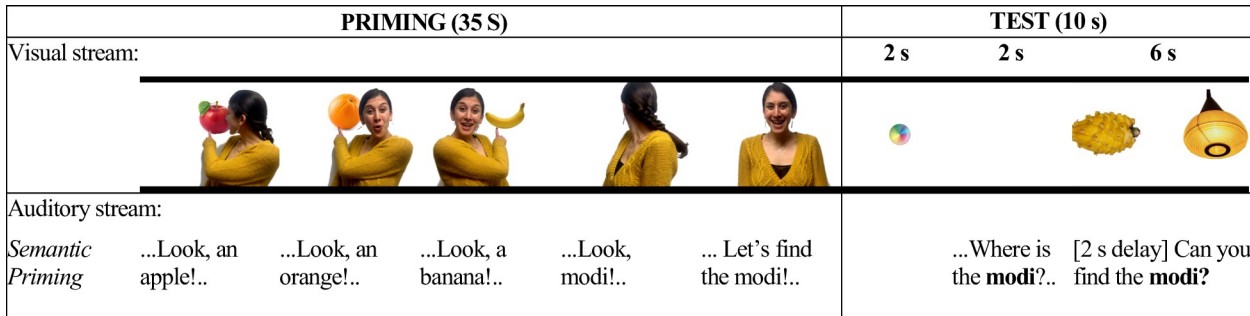

**Fig 1. A representative example of visual and linguistic information presented in each phase in the semantic priming condition.**

apple! Do you see the apple?"). A fourth object, seemingly unintentionally occluded by the actor's body, will be labeled with a novel noun (e.g. "Look, a *modi*!"), creating a situation of abstract reference—during naming, no referent object is visible. The actor will alternate between looking to the object and to the camera to indicate that the referent object is located behind her. We present infants with familiar word-object pairs during the priming phase to access their resident semantic knowledge [30,37–39].

Test (10 s): During Test, infants first will see an attention getter (2 s), followed by a blank white screen (2 s), during which they will hear the first prompt to look to the object corresponding to the novel label, e.g., "Now look! Where is the modi?". Immediately after, two objects will appear side-by-side—one is a member of the same semantic neighborhood during the Priming phase (e.g., a novel fruit) and the other is a semantically distant item (e.g., a novel furniture item). With these objects visible, infants will hear, e.g., "Can you find the modi?".

No Priming condition (*N* = 24; Fig 2): The procedure will be identical to the Semantic Priming condition, except that during Priming, infants will be presented with familiar word-object pairs, all from different semantic neighborhoods (e.g., a hat, truck, and horse). Thus, no priming of any particular semantic neighborhood will be involved.

Follow-up Control condition (*N* = 24): Although our pilot data (see below) suggest that 15-month-olds successfully establish novel object representations based on language input alone, it is important to rule out the possibility that their success is driven by the semantic relations among the objects themselves and therefore not dependent upon infants' inferences about novel word meanings. We therefore plan a Follow-up Control condition, identical to the Semantic Priming condition with one exception: during Test, infants will hear a novel word that differs from that presented on Trial 4. If infants' object preference at test is driven by the objects themselves, rather than naming, then performance in this Follow-up Control should not differ from that in the Semantic Priming condition. Conversely, if performance in the Semantic Priming condition is a consequence of verbal reference, then performance in this Follow-up Control condition should not differ from that in the No Priming condition.

**Data preparation.**   We will analyze infants' looking behavior from the onset of the image presentation through the end of the trial (6 s). Our decision to conduct 6 s of image presentation in test trials is based on prior evidence: Ferguson et al. (2014) [27] tested infants of comparable age in a procedure that also involved establishing word meanings in the absence of a visible referent. As in Ferguson et al., trials on which the total looking to the screen is less than 1s will be excluded from analyses.

For data collected in the lab using a Tobii T60XL eyetracker, we identify areas of interest (AOI) of approximately 720*790 pixels around each of the object images, such that every AOI includes a 50-pixel-wide area around an object image. This is done to accommodate for

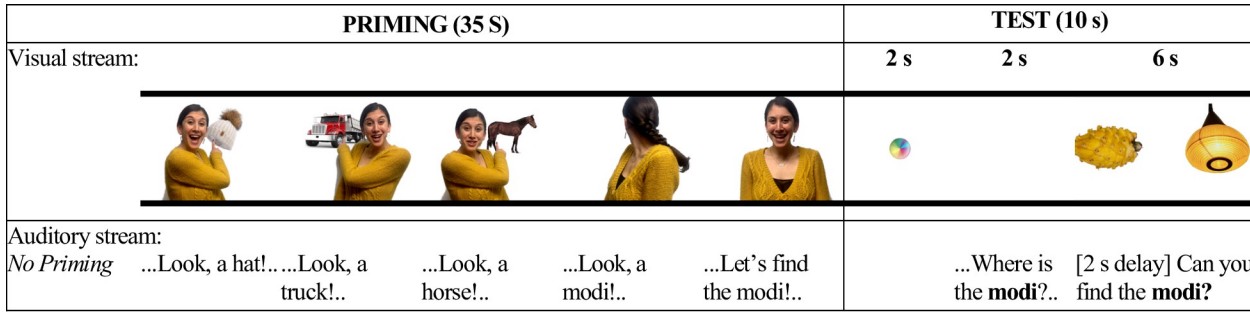

| **PRIMING (35 S)** | | | | | **TEST (10 s)** | | |
|---|---|---|---|---|---|---|---|
| Visual stream: | | | | | **2 s** | **2 s** | **6 s** |
| | | | | | | | |
| Auditory stream: | | | | | | | |
| *No Priming* | ...Look, a hat!.. | ...Look, a truck!.. | ...Look, a horse!.. | ...Look, a modi!.. | ...Let's find the modi!.. | ...Where is the **modi**?.. | [2 s delay] Can you find the **modi?** |

**Fig 2. A representative example of visual and linguistic information presented in each phase in the no priming condition.**

imprecisions of the eyetracking data, which often result from infants' head movement or blinking. Gazes outside these areas will be excluded from analysis, as will be trials in which an infant's total looking time is less than 1 out of 6 seconds.

For data collected on Lookit, manual frame-by-frame coding will be performed by trained research assistants who will watch video-recordings of online testing sessions and determine whether the infant is looking to the left or to the right side of the screen during each frame. Depending on the location of the target object, left-right codes will then be converted to target-distractor codes by a different research assistant. Ten percent of all videos will be selected for reliability coding. All research assistants involved in video coding will be agnostic to experimental conditions or counterbalancing schemes.

For each trial of each infant, we will calculate the proportion of looking time devoted to the target [looking to target/(looking to target + distractor)] throughout the analysis window. This should yield up to 4 data points for each infant (one per each of 4 trials). To assess the time course of infants' attention to the animal and artefact images, eyetracker data will be aggregated by condition into a series of 100-ms bins. For data collected on Lookit, we will use 200-ms bins. Web-cameras typically have sampling rates varying between 30HZ and 60HZ, which translates to frame duration ranging from 16.67 ms to 33.33 ms. Binning our data into 200-ms segments allows us to have at least 6 frames in each bin and to calculate the proportion of infants' looking to the target image. Fewer frames per bin would result in substantially cruder estimations of infants' proportion of looking to the target.

**Analyses.** The *dependent variables* (DVs) will be [1] infants' proportion of looking time (LT) to the target image from the image display onset until 6 s later, averaged across four trials; [2] the time course of the proportion of LT for each 100-ms (or 200-ms) bin in the 6-s window. Because our goal is to evaluate the effect of semantic priming on infants' word learning under the conditions of displaced reference, we will use Condition as the main *independent variable* (IV).

In analyzing the overall proportion of LT to the target, we will use a Generalized Linear Mixed Model (GLMM) with Condition as a fixed factor and Participant and Test Item as a random factors. We include the Test Item variable to account for the potential effects of infants' knowledge in different domains–fruits, vehicles, clothing, and animals. We plan to use an identity link, which will require that we test the distribution of the data for normality. We will implement this by running a Shapiro-Wilk test. If the distribution of our data significantly deviates from normal, an appropriate transformation will be applied. In analyzing the time course, we will use the cluster-based permutation analysis [40] to identify significant divergences between the conditions that correspond to mention of the target word.

The effects of demographic factors, vocabulary scores, and trial order will be tested in a preliminary analysis, implemented by fitting a GLMM to the DVs. Trial order is included to test whether infants' looking preferences change over the duration of the procedure, as they might lose interest or exhibit a learning effect as the experiment progresses. Only the factors that produce significant effects in the preliminary analysis will be included in the main analyses. No other analyses are planned for the data collected via the demographic survey.

**Predictions.** Because infants as young as 12 months infer the presence of a hidden object by following a speaker's line of regard [41], understand that words communicate about mental states [42–44], and appreciate referential cues [5], we make the following predictions.

**Proportion of LT to the target.** <u>Semantic Priming condition:</u> We predict significant effects of Condition, with average proportion of LT (across 6 s) to the target significantly higher in the Semantic Priming than in the No Priming and Follow-up Control conditions. Further, we predict that if infants successfully infer the meaning of the novel word and establish a representation of its referent in the Semantic Priming condition, their LT to the target will be significantly above chance (50%).

No Priming condition: If infants' looking behavior is influenced by semantic inferences, rather than baseline preferences, then their LT to both objects in the No Priming condition will be equivalent and will not deviate significantly from chance. We predict this because there is no basis for infants to infer the meaning of the novel word after being presented with three semantically distant word-object pairs.

Follow-up Control condition: If infants' looking preference in the Semantic Priming condition is the result of the semantic relations among the objects themselves, then their performance in the Follow-up Control condition will not differ from the Semantic Priming condition. Conversely, if infants' looking in the Semantic Priming condition reflects infants' inferences about novel word meanings, their LT in the Follow-up Control condition should not significantly deviate from chance.

## Time course

We expect that the time course data will show significant divergences between the conditions after each verbal prompt. Based on our pilot data, we envision that after each prompt, infants in the Semantic Priming condition will exhibit increased proportion of looking to the target, which we predict to be above the chance level. In the Follow-up Control condition, infants' looking behavior is likely to not be significantly affected by the verbal prompts because according to our hypothesis, they will have no particular expectations about the meaning of the novel word. It is, however, also possible that they will exhibit a weak preference for the target object (the object that is target in the Semantic Priming condition) based on their expectations about the next item shown given the relatedness of the three demonstrated objects. In the No Priming condition, infants have no basis to form any particular expectation about the object or the meaning of its label. For this reason, we expect that infants' looking behavior will not be affects by the verbal prompts and remain around the chance level for the duration of the test trial.

**Pilot data.**    The results of the pilot data analyses ($N = 18$) in the Semantic Priming condition reveal a significant looking preference for the target. Infants' LT to the target image across 6 s was significantly above chance (50%), $M = 64\%$, $SD = 8\%$, $t(17) = 5.5$, $p < .001$. The time course data reflects infants' reactions to verbal prompts: infants' looking to the target increased after each mention of the target word (Fig 3). These data provide assurances that the task is well-suited to test 15-month-olds' ability to learn novel word meanings from displaced reference and provides hints that infants might indeed be using semantic priming to infer such meanings.

## Results and discussion

We reasoned that if 15-month-olds can establish novel word-object mappings from language alone, it would indicate that they *can resolve abstract verbal reference*. Alternatively, failure on this task will raise several possible interpretations. First, 15-month-old infants may be sensitive to semantic priming [29], but not sufficiently robust to support learning a novel word in the absence of its referent. Second, infants may fail to infer that there is an object behind the actor. If this is the case, further investigation will be necessary to clarify the presence of a hidden object. Third, it is possible that 15-month-olds can resolve abstract verbal reference, but that their exposure to the novel word during priming is insufficient for them to establish novel word-object mappings from displaced reference. These possibilities all warrant follow-up investigations, which will shed light on the development of infants' inferential and representational capacities and their ability to use such capacities in word learning. We have designed such follow-up studies and will conduct them if 15-month-olds fail to learn the meanings of novel words in Experiment 1. However, based on the pilot data, we predict that these infants will succeed, which leads us to Experiment 2.

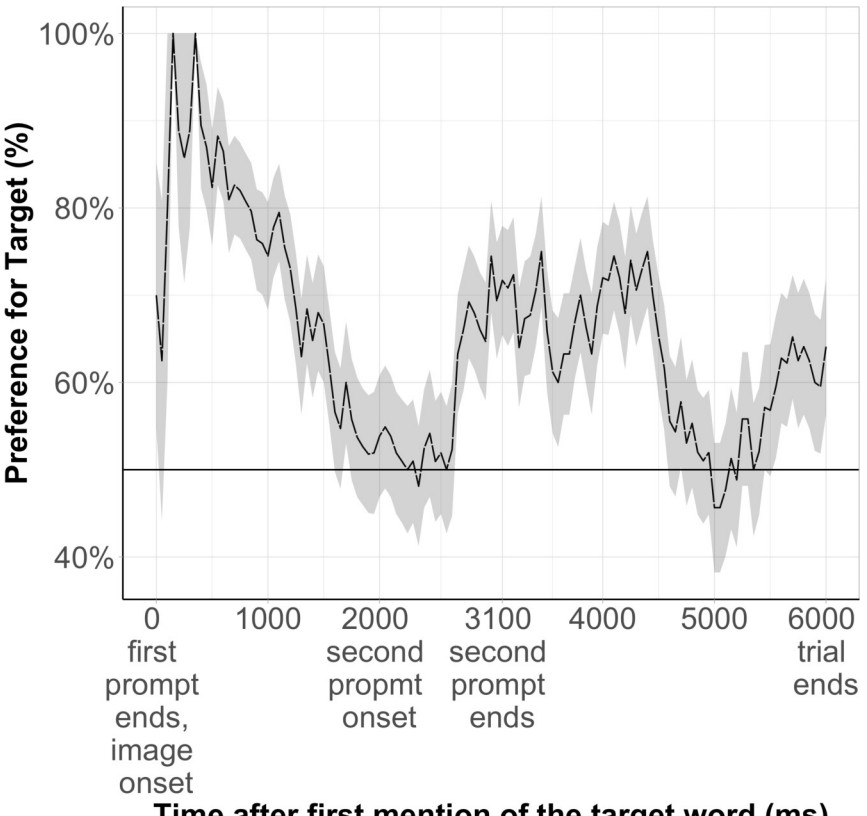

**Fig 3. Time course of the pilot data ($N$ = 18) in the semantic priming condition.**

To be expanded when data collection and analyses are complete.

## Experiment 2

If the results of Experiment 1 suggest that at 15 months, infants successfully establish novel word-object mappings from language alone, then we will go on to identify entry points for this capacity. More specifically, we will focus on 12-month-old infants, whose command of representational [43–45], lexical [46], and communicative capacities [5,47] suggests that they may be able to successfully resolve abstract verbal reference. By at least by 12–13 months, naming profoundly influences infants' learning. Applied consistently, words help infants form conceptual categories [45,48,49], as well as guide their attention to commonalities among and differences between individual exemplars [50]. Further, infants as young as 13-months-old rely on labels to infer non-visible properties and extend them to novel instances of a category, even when they are not especially perceptually similar to the learning exemplars [e.g., 51,52]. These early capacities lead us to hypothesize that provided with a large enough lexicon, 12-month-olds may successfully learn novel words from language input alone. Experiment 2 investigates this possibility.

Focusing on 12-month-olds, however, raises a new challenge. For our semantic priming manipulation to succeed, infants must comprehend at least 3 words from each semantic neighborhood under investigation (e.g., fruits, clothing, vehicles). Yet 12-month-olds' existing lexicons are rather sparse: only 30% of 12-month-olds meet this lexical requirement [28]. To redress this limitation, we initiate an at-home vocabulary training phase designed to bolster infants' lexical comprehension before they participate in the experiment proper.

## Method

**Participants.** Participants will be 72 infants ($N$ = 24 per condition: Semantic Priming, No Priming, Follow-up Control) recruited from Chicagoland area. All recruitment and consent procedures will be identical to Experiment 1. Participants may be excluded and replaced due to fussiness, crying, or equipment failure.

Apparatus, stimuli, and data preparation will be identical to Experiment 1. All IVs, DVs, and analyses will also be the same as in Experiment 1, except that infants' LT to the target on the familiar word knowledge test will be added to the main analyses.

**Timeline.** We intend to begin data collection as soon as the current registered report is granted *in-principle acceptance*. We expect that all data will be collected and analyzed within a calendar year from the completion of Experiment 1. The projected timeline to complete both proposed experiments is two calendar years from the *in-principle* acceptance date.

**Procedure.** The experimental procedure will be identical to Experiment 1 with one exception: before infants participate in the experiment proper, their parents will be enlisted to provide an at-home vocabulary training phase to bolster their infants' lexical comprehension. Two weeks before their lab visit (or online participation on Lookit), participants' parents will receive an online digital picture book containing 24 word-image pairs, including the 12 pairs featured in the Priming phase of the experiment proper. The order of word-image pairs will be randomized among participants. Parents will be asked to read their book at least once per day for 14 days, pointing to each picture and articulating its name. Before parents begin reading the book to their infants, they will view a video demonstration that will provide guidelines to reading this book to their infants and suggestions for establishing a daily reading routine (e.g., read before bedtime or after the nap, etc.). Moreover, parents will be contacted (email or text, as the parent prefers) every other day, reminding them to read the book to their infant. Parents will also be asked to fill out an online form daily (see S4 Appendix), reporting how engaged their infant was during the reading sessions. This at-home training should bolster infants' lexical knowledge, permitting us to test their ability to learn words from displaced reference when they visit the lab.

When infants visit the lab (or participate in the study online via Lookit), we first test comprehension of the 12 words included in the experiment proper. On each of 12 trials, infants will see two images from their book—one that will be used in Priming (e.g., apple, a target) and another (e.g., spatula, a distractor). Following Bergelson and Swingley [2012; 53], infants will be prompted to look at the target ("Look, an apple!"). The proportion of LT to the target will be evaluated against chance (50%). It is possible that infants will fail to learn words used in Priming during the two-week exposure to picture-book reading. If this is the case, we will explore alternative methods of boosting infants' word knowledge and consider alternative way of testing their ability to establish new word-object mappings from abstract reference.

**Predictions.** We expect that on the comprehension task that tests infants' knowledge of the familiar 12 words, infants will exhibit excellent performance and look to the target images over 70% of the time over the course of the trial. In the experiment proper, if 12-month-olds can indeed establish a representation of word meaning from abstract verbal reference, they should successfully map the novel word to the unseen object, echoing the results of Experiment 1. We expect to observe a significant effect of Condition on both DVs–the overall proportion of LT to the target across 6 s and the time course of infants' LT. We predict that the direction of differences will be the same as in Experiment 1. Further, if our hypothesis is correct, then infants' performance on the comprehension test will be correlated with their success in the experiment proper. However, if despite vocabulary training, 12-month-olds fail to learn novel words, this will either suggest that these infants cannot yet resolve abstract verbal

reference, or that their sensitivity to semantic priming is insufficiently robust to guide inferences about absent objects. Additional investigations will be required to distinguish between these possibilities.

## Results and discussion

To be added when data collection and analyses are complete.

## General discussion

Our goal of this paper was to fill the gap about the development of abstract verbal reference and address the outstanding questions about younger infants' ability to learn novel words from language input alone. To be expanded when data collection and analyses are complete. Implications for the co-development of language and cognition and the origins of the human ability to extract information from symbolic input will be discussed.

## Supporting information

**S1 Appendix. Augmented MacArthur short form vocabulary checklist: Level II.**
(PDF)

**S2 Appendix. Demographic questionnaire.**
(PDF)

**S3 Appendix. Visual* and auditory stimuli in Experiments 1 and 2.** *Source of images in the Test phase: https://unsplash.com.
(PDF)

**S4 Appendix. Vocabulary training tracking form.**
(PDF)

## Acknowledgments

We thank Caitlin Draper, Courtney Goldenberg, Rachel Kritzig, Ren Mondesir, Kiki Ogbuefi, Mary Okematti, Katelyn Pass, Judith Roeder, Annalisa Romaneneko, Murielle Standley, and Victoria Vizzini for help with pilot data collection. We also thank Miriam Novack and Alexander LaTourrette for their contribution to stimuli creation, helpful discussion about the study design, and help with pilot data analyses.

## Author Contributions

**Conceptualization:** Elena Luchkina, Sandra R. Waxman.

**Data curation:** Elena Luchkina.

**Formal analysis:** Elena Luchkina.

**Funding acquisition:** Sandra R. Waxman.

**Investigation:** Elena Luchkina, Sandra R. Waxman.

**Methodology:** Elena Luchkina, Sandra R. Waxman.

**Project administration:** Elena Luchkina, Sandra R. Waxman.

**Resources:** Sandra R. Waxman.

**Software:** Elena Luchkina.

**Supervision:** Sandra R. Waxman.

**Visualization:** Elena Luchkina.

**Writing – original draft:** Elena Luchkina, Sandra R. Waxman.

**Writing – review & editing:** Elena Luchkina, Sandra R. Waxman.

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
