## [Decision Letter · Decision Letter 0]

10 Aug 2020

PONE-D-20-14424

Category priming promotes infants’ success in imagining and naming things unseen

PLOS ONE

Dear Dr. Luchkina,

First of all, thanks for your patience waiting for this decision. As I mentioned in a previous email, it was a challenge to find reviewers during this challenging time, and the timing of when those reviews came in delayed my decision.  After careful consideration, we feel that it your submission merit but does not fully meet PLOS ONE’s publication criteria as it currently stands. Therefore, we invite you to submit a revised version of the manuscript that addresses the points raised during the review process.

I think this is an interesting proposal, and with some additional details and shifts in framing, as well as some possible modifications of some of the methodological details, could make a nice contribution to the literature on word learning. I won't go into detail with regard to the reviewers' comments, but let me highlight a few key points that I want to make sure you pay particular attention to:

1) Both reviewers thought that the theoretical motivation for these studies needed to be fleshed out, and had suggestions for how to do so. Reviewer 2 in particular was unclear what question this was asking that was distinct from prior work on word learning and learning about absent referents, and suggested that the work needed to be more clearly situated in the word learning literature. 

2) Both reviewers had some concerns about specific aspects of the methodology, including sample size, the possible need for an additional control condition, and more details about and justifications for specific methodological decisions.

3) Reviewer 2 notes, and I agree, that you may need to discuss how data collection will be impacted by the current pandemic. Is the plan to wait until after the pandemic subsides (which could be a long time), or are there plans for virtual data collection, in which case is there evidence that this will work with infants of this age? 

Please also be sure to read each review closely, as my above summary is by no means an exhaustive list of the recommended changes. 

We look forward to receiving your revised manuscript.

Kind regards,

Lucas Payne Butler

Academic Editor

PLOS ONE

Journal Requirements:

2. There are several instances where placeholders have been inserted into the text, to be completed once results have been obtained, e.g. " To be expanded when data collection and analyses are complete.".

Please note that, should the article be accepted, the submitted Protocol will be published as an article that will be seperate from the final Research Article that includes results.

Please revise the text accordingly, keeping in mind that the Registered Report Protocol should be considered as a standalone article.

For further details please see here: https://journals.plos.org/plosone/s/other-article-types#loc-registered-reports.

4. We note that Figures 1 and 2 in your submission contain copyrighted images.

All PLOS content is published under the Creative Commons Attribution License (CC BY 4.0), which means that the manuscript, images, and Supporting Information files will be freely available online, and any third party is permitted to access, download, copy, distribute, and use these materials in any way, even commercially, with proper attribution. For more information, see our copyright guidelines: http://journals.plos.org/plosone/s/licenses-and-copyright.

a. You may seek permission from the original copyright holder of Figures 1 and 2 to publish the content specifically under the CC BY 4.0 license.

Reviewers' comments:

Reviewer's Responses to Questions

**Comments to the Author**

1. Does the manuscript provide a valid rationale for the proposed study, with clearly identified and justified research questions?

Reviewer #1: Yes

Reviewer #2: Partly

2. Is the protocol technically sound and planned in a manner that will lead to a meaningful outcome and allow testing the stated hypotheses?

Reviewer #1: Partly

Reviewer #2: Partly

3. Is the methodology feasible and described in sufficient detail to allow the work to be replicable?

Reviewer #1: Yes

Reviewer #2: Yes

4. Have the authors described where all data underlying the findings will be made available when the study is complete?

Reviewer #1: Yes

Reviewer #2: Yes

5. Is the manuscript presented in an intelligible fashion and written in standard English?

Reviewer #1: Yes

Reviewer #2: Yes

6. Review Comments to the Author

You may also provide optional suggestions and comments to authors that they might find helpful in planning their study.

Reviewer #1: This is a well written manuscript, with clear hypothesis. The design is clever, thorough and deals with potential confounds, thus allowing a straightforward validation of hypothesis. However, some additional clarifications are required and some additional confounds addressed.

The intro should discuss evidence for superordinate category knowledge in the second year of life. Your experiment depends on children being able to recognise a new fruit as belonging to the fruit category, for example. More detail needed also on the prerequisites for the semantic priming - The study you cite in support of such an ability (19) only shows that younger infants have broader categories, e.g. foot includes socks - it is not a study suggesting semantic priming. The developmental change described in that study might suggest that 15 mo do not even benefit from this over-generalization. Also, is priming sufficient, do infants not also need to make a pragmatic inference - that since the experimenter choose to name 3 fruits, the fourth must be from the same superordinate category? This is quite a peculiar inference which also needs to be backed up with some theoretical or empiric support from the literature. You preliminary study shows evidence that either semantic priming or pragmatic inferences do work in 15mo, so what is needed is some stronger apriori motivation of why it might be so.

Please provide power analysis for your sample - how many participants are needed to reach significance for both the main effect of Condition and the follow-up tests, for the expected (or medium) effect size. Please ensure that recruitment takes into account drop-out rate, thus ensuring that the final sample size is that expected from power analysis. Also, indicate the follow-up tests and any multiple correction applied, as part of your analysis.

Follow-up control condition - Looking towards the target after the first prompt is not strong evidence that they have not associated the word with the target - participants may make the pragmatic choice of looking at the category that the person engaged with previously. A better test would be to use a completely novel word in test.

Experiment 2- related to the comment above, if the training is aimed at teaching superordinate category knowledge, will labeling basic level categories achieve that ? If it is assumed 12 mo possess the superordinate categories, do they need to know the basic level labels in order to succeed at the task ?

Reviewer #2: Discussion of Theoretical Framing of the Research Question

Overall this is a well written proposal. I have several concerns related to the framing of the study and the specific research question.

The introduction starts by stating that (1) calling to mind referents in their absence is a crucial feature of human language, (2) this ability is enabled by an understanding of the referential understanding of words, and (3) the developmental origin of this referential understanding is not clear.

The authors then discuss some literature on bootstrapping on word meaning from syntax, and state that 15-month-olds fail to use syntax as a cue to a word’s meaning and pose the question of whether 15-month-olds have the conceptual and linguistic knowledge to learn words from language alone. Their study is aimed to answer this question.

I found this set-up of the study questions very confusing for the following reasons:

1. There is plenty of evidence in the word learning literature that infants have an understanding of the referential nature of labels by 15-months. For example, work by Bergelson and Swingley shows that 6-9 month olds have a fairly abstract representation of labels; several studies by Ganea and Saylor show that 12-15 month-olds respond to labels in a displaced reference context; Ganea, Fitch, Harris and Kaldy (2016) show that even 15-month-olds can update expectations about the visual word on the basis of language alone; and, directly relevant to this study’s goals, a recent study by Osina, Saylor, and Ganea (2018) indicates that infants as young as 16-months use category label knowledge to interpret absent reference. Also, the findings of Gliga and Csibra (2009) indicate that infants expect to find an objects based on a novel label coupled with a deictic gesture. Given this body of work (this is only a small selection of it) showing that infants have a referential understanding of language by 15 months, as shown by their ability to process labels in the absence of their referents and their ability to access categorical knowledge when hearing a label, it is not clear to me what the authors define as “referential understanding.” The authors need to state explicitly why these other ways of testing referential understanding do not provide sufficient evidence and in what way their study is advancing what we already know about infants’ referential understanding.

2. The study should be better situated in the word learning literature. It is not clear to me why the work on syntactical bootstrapping is included. This study is not using syntax but conceptual knowledge of a category to see whether children can infer meaning. Therefore a a review of the literature on the taxonomic bias on word learning needs to be included, rather than of the syntactical bootstrapping literature.

Given the above considerations, it was difficult to fully comprehend the exact research questions. It seems like the goal is to test whether toddlers establish a referent when they hear something even if they do not see the object, when it is taxonomically related to previous objects in the task. The novelty seems to be that a specific referent (using a novel not familiar word) can be recalled when the category is primed. What is the nature of children’s expectations of what a novel word means? Can children learn words in the absence of information, and does the same taxonomic bias occur without visual information?

Discussion of the Design

- For the condition effects, there were only predictions for referential understanding when toddlers were primed by category. There should also be expectations for the no category priming condition (the basis for this expectation also needs to be supported by the literature and then cited accordingly).

- Since the goal of the proposed research is to test the influence of conceptual priming on infants’ referential understanding, the perceptual cues should be minimized/held constant within and across conditions. Some perceptual features of the current stimuli might be potential confounding variable in the proposed study. For example, in the first trial of the Category-priming condition, the novel fruit is both conceptually and perceptually similar to the preceding set of stimuli while the non-target item (furniture) is both conceptually and perceptually distinct. An out-of-category item which perceptually looks more similar to the target category would work better.

- Decisions for how the question for the control condition was selected need to explained. Why and how they help address the research question.

- Why 6 seconds? Explain why this amount of time was selected and if there is research to suggest that this is the norm. Also, the exclusion criteria for the looking data needs to be clear. What is the threshold criteria to be included for each trial, and how was this decided.

- Explain the sample size – power analysis

- Why did the authors select this type of design as opposed to a within-subjects design? They could use 2 trials per condition instead of 4 – increases the power.

- What will the authors do with the information collected from the demographic questionnaire? Will they link any of the data to the study’s DVs?

- The timeline for data collection may need to be adjusted given COVID situation.

7. PLOS authors have the option to publish the peer review history of their article (what does this mean?). If published, this will include your full peer review and any attached files.

Reviewer #1: **Yes: **Teodora Gliga

Reviewer #2: No

---

## [Author Response · Author response to Decision Letter 0]

28 Oct 2020

Dear Dr. Butler,

Please consider our revised manuscript PONE-D-20-14424 for publication in PLOS ONE. We are grateful to you and the reviewers for your thoughtful feedback. We hope that you agree that by responding to this very productive set of reviews, we have produced a significantly stronger manuscript. Please see below point-by-point responses to reviewer’s comments. 

Sincerely,

Elena Luchkina

Summary of the major changes:

1. We clarified that the central construct under investigation is “abstract verbal reference”, which we define operationally as “an appreciation that words are linked to mental representations that can be established, retrieved and modified, even when the entities to which they refer are not perceptually available”. See §1 of the Introduction, as well as §2 about “displaced/fragmented reference” as its operationalized forms. 

2. Following R1’s and R’s excellent suggestion, we no longer frame the work in the literature on superordinate category knowledge. We now focus on semantic neighborhoods as our main experimental manipulation. We have changed the title to reflect this modification and renamed the conditions from Category Priming and No-Category Priming conditions to Semantic Priming and No Priming conditions respectively.

3. Based on R1’s suggestion, we redesigned the Follow-up Control condition. It now addresses the possibility that success in the Semantic Priming condition is due to semantic relations among the objects themselves and is not dependent upon infants’ inferences about novel word meanings.

4. Per R2’s suggestion, we expanded the introduction to better situate the current project within the word-learning and reference literatures. 

5. We also modified the test stimuli included in all experiments and conditions to address R2’s suggestion to minimize perceptual distance between target and distracter images.

Reviewer 1

The reviewer suggested that the test for our hypothesis would be stronger if we do not to locate our predictions in relation to superordinate category knowledge in infants this young. 

We are grateful to R1 for the suggestion. We agree that we need not rely upon superordinate category knowledge to test abstract verbal reference. Instead, we frame the hypothesis in the context of the semantic priming available from known words. We propose that the ability to establish a representation of a hidden referent based on a semantic neighborhood would constitute strong evidence of resolving abstract reference (see p. 6, §1).

The reviewer also raised a concern that infants’ looking behavior in the Semantic Priming condition (formerly “Category Priming condition”) could be driven by the relations between the objects rather than by infants’ inferences about the meaning of the novel word. 

We thank the reviewer for bringing this up. In response, we changed the design of the Follow-up Control condition, which now addresses this concern. Now, the Follow-up Control condition is designed identically to the Semantic Priming condition, except that infants hear an unfamiliar novel noun during test trials (see p. 10, §1). This way, we can discern between the effects of the relation between the objects and infants’ semantic inferences about the novel word. 

The reviewer asked that we provide power analysis for our sample, taking into account the drop-out rate, and indicate what multiple corrections would be applied.

We now added a power analysis and a recruitment plan that incorporates the projected drop-out rate (p. 6, §2). Because our planned analyses are based on GLMM, which (as we now point out) includes Test Item as a random factor, we do not need additional multiple corrections.

The reviewer suggested that we use a completely novel word during Test in the Follow-up Control condition to evaluate whether infants associated the word with the target object.

As mentioned above, we have done so. (p. 12, §1). 

Reviewer 2

The reviewer asked that we ground our work more comprehensively in the literature on word learning and outline the new insights that our work will bring to the problem of verbal reference. 

We appreciate this suggestion and have done so, now including the references cited by the reviewer and many others (pp. 3-4). We highlighted the difference between the extant empirical investigations and the current proposal, as well as our contribution to this literature (p. 4, §2; “What remains unknown is when infants successfully acquire new representations and update them on the basis of language alone.”). 

The reviewer also suggested we define more clearly the construct under investigation.

We appreciate this suggestion. We clarified that the central construct under investigation is “abstract verbal reference”, which we define on p. 1, §1. 

The reviewer asked that we explicitly state our predictions for each condition.

We have now added explicit predictions for each condition (pp. 12-13).

The reviewer suggested that we minimize the perceptual differences between test images within and across conditions. 

We appreciate this suggestion, and have edited the test stimuli accordingly (see Appendix 3).

The reviewer asked us to clarify the advantages of the Follow-up Control condition, pointing out how they help address the research question.

We appreciate this suggestion. In response, we have clarified that the Follow-up Control condition (which now has been modified following R1’s suggestion; see p. 10, §1) will help discern between the effects of the relation between the objects and infants’ semantic inferences about the novel word. 

The reviewer suggested that we clarify how we decided on the duration of test trials and that we outline exclusion criteria for the looking data. 

We now articulate the reasons for 6 s test trials and the threshold criteria in the pilot and proposed studies. Both are based on earlier investigations of infants’ ability to take advantage of existing word knowledge to infer the meaning of novel words (p. 10, § 2; ). 

The reviewer asked that we explain the sample size and provide power analysis.

We now added power analysis on p. 6, §2.

The reviewer asked that we clarify why we chose to a within-subjects design with 4 test trials.

We have done so on p. 8, §2.“We selected a between-subject design to avoid potential carry-over effects, which would dilute the predicted advantage of Semantic Priming”.

The reviewer also asked us to explain to we would analyze the information collected from the demographic questionnaire and whether it would be linked to the DVs.

The demographic data will used in preliminary analyses and may be included in the main analyses if the preliminary analyses show a significant effect. We do not have specific hypotheses about the role of the demographic factors in the development of abstract verbal reference comprehension. We clarify this on p. 12, §2.

The reviewer suggested that we provide a COVID-19 contingency plan and outline an adjusted timeline for data collection.

We added a COVID-19 contingency plan and explained the projected timeline of data collection on p. 7, §2.

---

## [Decision Letter · Decision Letter 1]

21 Dec 2020

Semantic priming supports infants’ ability to learn names of unseen objects

PONE-D-20-14424R1

Dear Dr. Luchkina,

We’re pleased to inform you that your manuscript has been judged scientifically suitable for publication and will be formally accepted for publication once it meets all outstanding technical requirements.

Kind regards,

Lucas Payne Butler

Academic Editor

PLOS ONE

Additional Editor Comments (optional):

Reviewers' comments:

Reviewer's Responses to Questions

**Comments to the Author**

1. Does the manuscript provide a valid rationale for the proposed study, with clearly identified and justified research questions?

Reviewer #2: Yes

2. Is the protocol technically sound and planned in a manner that will lead to a meaningful outcome and allow testing the stated hypotheses?

Reviewer #2: Yes

3. Is the methodology feasible and described in sufficient detail to allow the work to be replicable?

Reviewer #2: Yes

4. Have the authors described where all data underlying the findings will be made available when the study is complete?

Reviewer #2: Yes

5. Is the manuscript presented in an intelligible fashion and written in standard English?

Reviewer #2: Yes

6. Review Comments to the Author

You may also provide optional suggestions and comments to authors that they might find helpful in planning their study.

Reviewer #2: The authors have responded to all my suggestions. I look forward to seeing the results. Upon writing the paper I suggest the authors also cite this relevant work:

Saylor, M. M., Osina, M., Tassin, T., Rose, R., & Ganea, P. A. (2016). Creature feature: preschoolers use verbal descriptions to identify referents. Journal of Experimental Child Psychology. doi:10.1016/j.jecp.2016.07.005

7. PLOS authors have the option to publish the peer review history of their article (what does this mean?). If published, this will include your full peer review and any attached files.

Reviewer #2: No

---

## [Editor Report · Acceptance letter]

23 Dec 2020

PONE-D-20-14424R1 

Semantic priming supports infants’ ability to learn names of unseen objects 

Dear Dr. Luchkina:

I'm pleased to inform you that your manuscript has been deemed suitable for publication in PLOS ONE. Congratulations! Your manuscript is now with our production department. 

Kind regards, 

on behalf of

Dr. Lucas Payne Butler 

Academic Editor

PLOS ONE